# PvNeXt: Rethinking Network Design and Temporal Motion for Point Cloud Video Recognition

**Jie Wang[1], Tingfa Xu[1,†], Lihe Ding[2], Xinjie Zhang[3], Long Bai[2], Jianan Li[1,†]**
[1]Beijing Institute of Technology    [2]The Chinese University of Hong Kong
[3]Hong Kong University of Science and Technology
https://github.com/Roywangj/PvNeXt

## Abstract

Point cloud video perception has become an essential task for the realm of 3D vision. Current 4D representation learning techniques typically engage in iterative processing coupled with dense query operations. Although effective in capturing temporal features, this approach leads to substantial computational redundancy. In this work, we propose a framework, named as PvNeXt, for effective yet efficient point cloud video recognition, via personalized one-shot query operation. Specially, PvNeXt consists of two key modules, the Motion Imitator and the Single-Step Motion Encoder. The former module, the Motion Imitator, is designed to capture the temporal dynamics inherent in sequences of point clouds, thus generating the virtual motion corresponding to each frame. The Single-Step Motion Encoder performs a one-step query operation, associating point cloud of each frame with its corresponding virtual motion frame, thereby extracting motion cues from point cloud sequences and capturing temporal dynamics across the entire sequence. Through the integration of these two modules, PvNeXt enables personalized one-shot queries for each frame, effectively eliminating the need for frame-specific looping and intensive query processes. Extensive experiments on multiple benchmarks demonstrate the effectiveness of our method.

## 1 Introduction

Point cloud videos serve as a pivotal character, offering a dynamic perspective into our environment, which is fundamental in the realms of robotics and AR systems. These sequences, which present movements within the physical domain, are crucial in delineating environmental transformations and facilitating interactions within said environments. This contrasts starkly with the limited descriptive capabilities of 2D images or static 3D point clouds. Therefore, enhancing the ability of point cloud video perception becomes a significant yet challenging task. However, 4D data representation learning presents vital challenges and remains a nascent field of inquiry. The amalgamation of 3D geometry and dynamic motion often leads to data redundancy within an exceedingly high-dimensional space, which heavily hinders the development of efficient spatio-temporal representations.

Recent works (Fan et al., 2022; 2021b;a; 2023; Ben-Shabat et al., 2024; Huang et al., 2024) have attempted to overcome the limitation of 4D data processing through innovative architectures. Although effective in mining the temporal feature, these methods suffer from the same challenge: a tendency to extensively query features from neighboring segments in sequential frame point clouds for motion extraction, as seen in Fig. 2a. The accurate representation of video content heavily relies on temporal dynamics, which necessitates extensive computational efforts for the precise detection and analysis of dynamic motion in each frame. This process, especially in identifying motion correlations across successive frames, often results in significant computational redundancy.

By comprehensive scrutiny of existing pipelines for point cloud video understanding, we discover two intrinsic drawbacks of the current paradigm that decrease computational efficiency heavily. The

---

[†]Correspondence to: Jianan Li and Tingfa Xu.

first inherent flaw pertains to the constraints arising from the process of traversing each frame to extract local motion. Traditional methods of 4D representation learning typically involve iterating over each frame, querying points in proximate locations of neighboring frames, and capturing the motion variations in various local areas to apprehend the scene's dynamic information. Subsequent approaches in 4D representation learning have employed more sophisticated encoders (e.g., convolutional schemes (Fan et al., 2022; 2021b), or self-attention mechanisms (Fan et al., 2021a; 2023)) to extract finer geometric motion details. However, these methods still cannot avoid the traversal of all frames, leading to massive computational expenditure. The second inadequacy pertains to the substantial overlapping of the resultant local point frames acquired through grouping within the 4D space. This overlap results in a singular input point cloud video frame being encompassed by multiple point sets concurrently. Given that the embedding of point cloud frames is executed individually within each local point set, this scenario leads to the replication of point embeddings for the same point cloud frame across diverse point sets, we posit that this process requires noteworthy redundant computations.

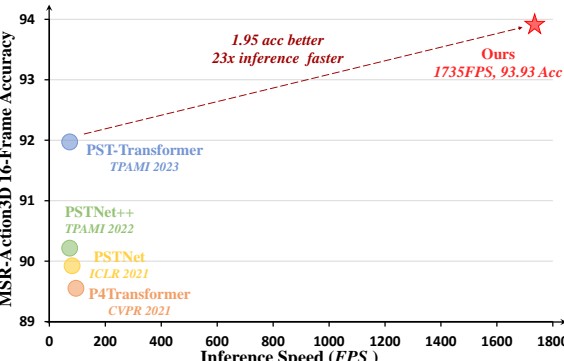

Figure 1: Comparisons about accuracy and inference speed between our algorithm and other baselines.

The aforementioned constraints lead us to explore a novel and efficient paradigm for point cloud video representation learning, guided by a principle of minimalist design. Our approach stems from a pivotal insight: frequent querying of adjacent frames may not be inherently linked to optimal performance and could inadvertently introduce superfluous computation. To address this, we eliminate the process of neighboring frame dense querying. This method relies solely on self-querying to extract local geometries, thereby significantly reducing computational costs. However, it presents a key challenge as it lacks the feature interactions of each frame, neglecting the motion between frames, which is crucial in capturing the geometric properties of temporal sequences.

In light of this, we introduce an additional step prior to the frame query process. This step implicitly links the motion information of points within specified local regions of adjacent frames, encapsulating their correlation and the motion of neighboring frames within local representations. Utilizing the learned motion, virtual frames are simulated. Each frame then only needs to query its corresponding virtual frame, as shown in Fig. 2b, inherently capturing the dynamic information of the video. This single-step query operation enables the network to achieve a natural spatio-temporal understanding of the video, while avoiding complex recurrent and dense query operations.

Building upon these key concepts, we have devised a novel architecture termed PvNeXt, tailored for the efficient analysis of point cloud videos. PvNeXt elucidates hierarchical features from the input point cloud video via the integration of multiple progressive learning stages. Each of these stages encompasses two distinct modules: the **Motion Imitator** and the **Single-Step Motion Encoder**. The former is responsible for capturing the temporal motions between selected frames and their subsequent frames. This module operates on a per-anchor basis, tracking the displacement of region centered on the sampled anchor, across consecutive frames. It effectively models the movement patterns within the point cloud. Subsequently, the Single-Step Motion Encoder module leverages the learned motion from the Motion Imitator to generate virtual frames. It performs a one-step query operation, mapping points from the original frame to their corresponding synthetic frames. This operation extracts geometric features, ensuring the synthetic frames are consistent with the original input and capturing intricate spatio-temporal relationships. Through the integration of these two modules, PvNeXt enables personalized one-shot queries for each frame, effectively eliminating the need for frame-specific looping and intensive query processes.

The effectiveness of PvNeXt is validated through extensive experiments on diverse datasets such as MSR-Action3D (Li et al., 2010) and more challenging NTU-RGBD (Shahroudy et al., 2016). Our proposed PvNeXt achieves significant improvement in computation overhead and memory consumption, and establish state-of-the-art performance. As illustrated in Fig. 1, our framework attains a state-of-the-art level of accuracy while concurrently delivering remarkable efficiency in inference speed, indicating its effectiveness and efficiency. Notably, PvNeXt achieves a significant improve-

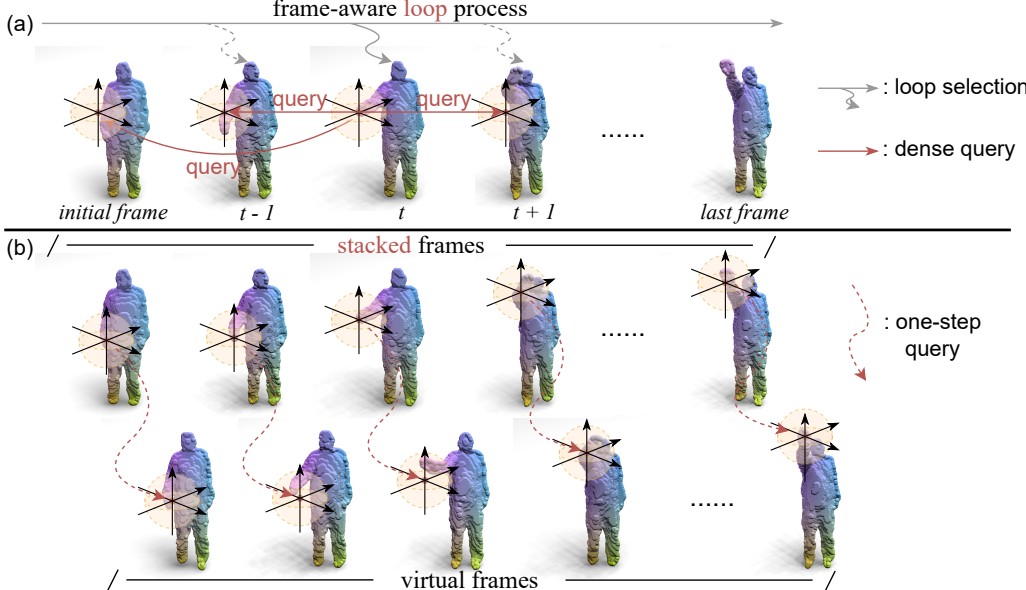

Figure 2: Illustration of various approaches to spatio-temporal modeling. (a) Current methods typically capture motion through iterative looping processes combined with dense query operations. (b) Our proposed method captures motion via personalized one-step queries targeted at virtual frames.

ment in performance on the widely utilized MSR-Action3D benchmark for point cloud video recognition, exhibiting a $1.95\%$ increase in accuracy alongside a remarkable $23\times$ speedup in inference compared to the PST-Transformer, with over $60\times$ fewer parameters. Our key contributions can be summarized as follows:

- We present a new framework, PvNeXt, which pioneers a novel and efficient paradigm tailored for point cloud video analysis.

- We propose a novel personalized one-shot query method, enabling the efficient spatio-temporal modeling for point cloud video.

- We demonstrate the effectiveness and efficiency of PvNeXt through extensive experiments on multiple point cloud video recogintion benchmarks.

## 2 RELATED WORKS

### 2.1 SUPERVISED POINT CLOUD VIDEO LEARNING

The comprehension of point cloud videos, as an extension of static 3D point cloud understanding, presents unique challenges due to the incorporation of complex spatial-temporal information. These videos possess a dual nature of being unordered (intra-frame) and ordered (inter-frame), thereby necessitating sophisticated approaches to aggregate and utilize spatial-temporal data for the perception of both geometry and dynamics in 4D point cloud sequences. Current methodologies in this domain can be primarily categorized into voxel-based and point-based techniques. Voxel-based approaches, as exemplified by MinkwoskiNet (Choy et al., 2019), involve the voxelization of raw point clouds followed by the extraction of features from 4D voxels using 4D convolutions. On the other hand, point-based methods, such as MeteorNet (Liu et al., 2019) and PSTNet (Fan et al., 2022), operate directly on raw points. MeteorNet, an extension of PointNet++ (Qi et al., 2017b), introduces a temporal dimension and performs explicit tracking of point motions for grouping. Similarly, PSTNet constructs a point tube along the temporal axis for 4D point convolution. Leading-edge techniques, including P4Transformer (Fan et al., 2021a), PPTr (Wen et al., 2022), and LeaF (Liu et al., 2023), fall within the point-based category and integrate the transformer architecture. This integration aims to obviate point-tracking and enhance the capture of spatio-temporal correlations. Building upon the foundation of P4Transformer, PST-Tranformer (Fan et al., 2023) further optimizes transformer utilization, with a specific focus on the temporal motion capture in point cloud videos. To address

the quadratic complexity of the transformer-based architectures, MAMBA4D (Liu et al., 2024) develops a novel generic 4D backbone for point cloud video understanding, which models long-range dependencies based on advanced State Space Models (SSMs) with linear complexity. Despite their efficacy, these methods involve substantial computational overhead due to the intricate handling of point cloud motion, including numerous loops and extensive query manipulation. This study introduces a novel algorithm that streamlines the process by eliminating the need for complex loops and query operations. Instead, our method efficiently learns motion through a simplified yet efficient one-step query operation, demonstrating a significant improvement in computational efficiency.

## 2.2 REPRESENTATION LEARNING ON POINT CLOUDS.

The effective extraction of discriminative features from point clouds is a crucial task in numerous 3D vision applications. Traditional methods, however, face challenges due to the intrinsic irregular nature of point cloud data. Conventional strategies, including voxel-based (Maturana & Scherer, 2015; Wu et al., 2015) and view-based (Guo et al., 2016; Qi et al., 2016; Su et al., 2015; Saha et al., 2022; Wiesmann et al., 2022) techniques, primarily attempt to reshape point clouds into more structured forms such as voxel grids or 2D images. These processes, though leveraging methodologies suited for structured data, often result in a diminution of information due to the projection involved. In response, point-based methods (Qi et al., 2017a;b; Wang et al., 2019) address these limitations by directly interacting with the raw point cloud data. PointNet (Qi et al., 2017a), a significant innovation in 3D data analysis, uses shared multi-layer perceptrons (MLPs) for learning distinct features at the individual point level. It preserves permutation invariance via a max-pooling operator. Extending this, PointNet++ (Qi et al., 2017b) refines this model by extracting both local and global geometric features through a hierarchical network structure and MLPs. Building upon this, PointNeXt (Qian et al., 2022) introduces an inverted residual bottleneck structure and separable MLPs to PointNet++, enabling efficient scaling of the model and optimizing performance. Additionally, PointMLP (Ma et al., 2022) presents a simple yet effective approach by integrating a feed-forward residual MLP with a geometric affine module, enhancing local feature extraction capabilities and offering robust representations of point cloud data. This paper introduces a novel, streamlined hierarchical learning framework designed specifically for dynamic point clouds, focusing on efficiently capturing local geometric motion, proposing an advanced paradigm for 4D point cloud representation learning.

## 3 METHODOLOGY

In point cloud video area, current methods tend to optimize the motion capture process via the extensive query operation from neighboring frames in sequential point clouds, leading to a superficial learning process that distracted by repeated adjuct information, failing to capture the depth and complexity dynamics in point cloud videos, as well as bring massive computational consumption. Instead, we propose an efficient paradigm that customizes each frame with personalized one-shot query, thus avoiding frame-aware loop processes along with the intensive queries.

### 3.1 PRELIMINARY: DENSE QUERY BASED POINT CLOUD VIDEO PERCEPTION PARADIGM

The design of dense-query-based methods for point cloud video analysis dates back to the PSTNet and P4Transformer (Fan et al., 2022; 2021a), if not earlier. The primary motivation behind this direction is to implicitly capture the dynamic of the point cloud video by identifying and associating relevant points within the surrounding spatio-temporal domain.

Given a set of temporal points $\mathcal{P} \in \mathbb{R}^{N \times T \times 3}$, where $N$ indicates the number of points in Cartesian space and $T$ denotes the frames of the input video, current methods (Fan et al., 2022; 2021b;a; 2023) aims to learn the spatio-temporal structures of $\mathcal{P}$ using customized 4D convolution operators.

One of the most pioneering works is PSTNet (Fan et al., 2022), which learns spatio-temporal structure through massive temporal looping and dense query operation in stacked multiple learning stages. In each stage, $N_s$ points are re-sampled by the farthest point sampling (FPS) algorithm where $s$ indexes the stage. Subsequently, the loop operation will be applied across the temporal dimension. When stepping into the $t$-th loop, each sample point will select $K$ neighboring points from frames within the temporal interval $\Delta t$ of current frame $t$. In frames within the temporal interval, $K$ neighbors are aggregated by max-pooling to capture local structures. Conceptually, the kernel

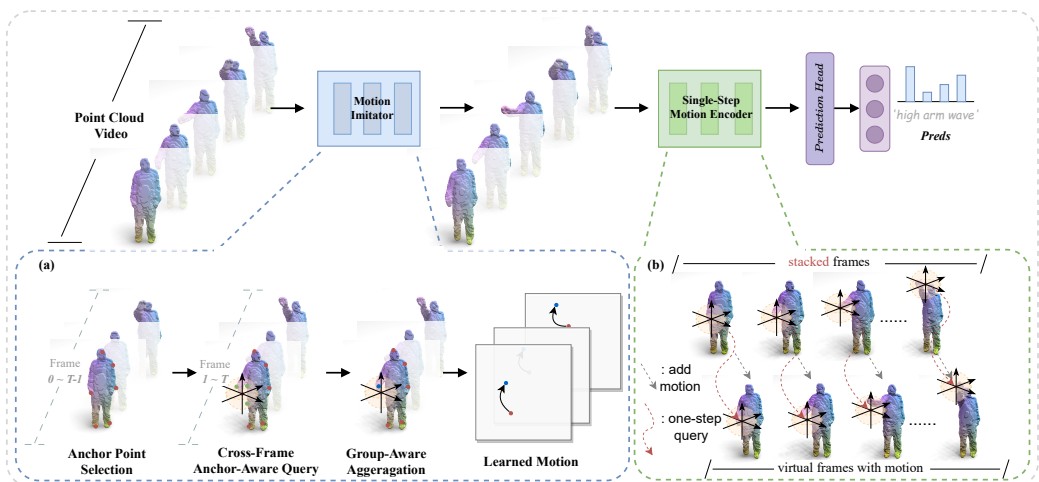

Figure 3: **Overall architecture** of the proposed one-step query PvNeXt workflow, composed of two key modules: (i) the Motion Imitator, which captures the motions between selected frames and their subsequent frames for each sampled point; and (ii) the Single-Step Motion Encoder, which utilizes the learned dynamics to generate synthetic frames and performs a one-step query from the original frame points to their corresponding synthetic frames to extract geometric features.

operation of can be formulated as:

$$g_i^{(\tau)} = \mathcal{A}\left(\Phi\left(f_{i,j}^{(\tau)}\right)\Big|j = 1, ..., K\right),\tag{1}$$

where $\mathcal{A}(\cdot)$ denotes a symmetric function, *e.g.*, max pooling, to aggregate encoded point features. $\Phi(\cdot)$ denotes the local feature extraction function, implemented by multi-layer perceptron (MLP). $f_{i,j}^{\tau}$ is the $j$-th neighbor point feature of $i$-th sampled point, where neighbors are from the $\tau$-th frame while the sampled point is from the $t$-th frame. Here, $\tau$-frame is a frame within the temporal interval of $t$-th frame, $t - \Delta t \le \tau \le t + \Delta t$.

Through such operations, PSTNet acquires the geometric features of all frames within the temporal sampling interval. Subsequently, PSTNet performs a symmetric function, *e.g.*, max pooling, along the temporal dimension, thereby implicitly extracting the motion within the frames.

$$g_i = \mathcal{A}\left(\left(g_i^{(\tau)}\right)\Big|t - \Delta t \le \tau \le t + \Delta t\right),\tag{2}$$

The aforementioned dense query step is executed when the iteration reaches frame $t$. By repeating the iterative process along with the dense query operation, the network is able to progressively enlarge the receptive temporal fields, capturing the motion across the point cloud video.

While existing methodologies effectively leverage intricate spatio-temporal data, yielding impressive outcomes, they encounter two principal challenges that inhibit further advancement. Firstly, the introduction of advanced spatio-temporal feature extractors significantly escalates computational complexity, thereby resulting in untenable inference latency. Secondly, with the development of delicate spatio-temporal convolution operators, the performance gain has started to saturate on popular benchmarks. Both limitations encourage us to develop a new method that circumvents the employment of sophisticated 4D feature extractors, and provides gratifying results.

## 3.2 Framework of PvNeXt

In order to get rid of the restrictions mentioned above, we present a simple yet effective network for point cloud video analysis that without any sophisticated loop process or dense query operations.

The proposed one-step query PvNeXt workflow is presented in Fig. 3, consisting of two key modules: i) the Motion Imitator, responsible for capturing the motions between selected frames and their next frames, specific to each sampled point; ii) The Single-Step Motion Encoder, by utilizing the learned dynamics, generates synthetic frames. Subsequently, it conducts a one-step query operation from points in the original frame to their corresponding synthetic frames to extract geometry.

### 3.2.1 MOTION IMITATOR

The Motion Imitator takes point cloud $\mathcal{P}$ as input, captures region motion and enables multiple region geometric shift learning. To this end, we first select $M$ anchor points from $\mathcal{P}$, query neighbor points from the next frames centered on these anchor points, then aggregate local features and establish the inner connections between them, finally learn group-aware motion.

**Cross-Frame Anchor-Aware Query.** In this step, a set of $M$ anchor points $\mathcal{D} \in \mathbb{R}^{M \times T \times 3}$ is initially selected from the input point cloud video $\mathcal{P}$, via the Farthest Point Sampling (FPS) algorithm. These anchors form the foundation for the subsequent anchor-wise motion learning process, wherein each anchor undergoes independent transformation.

For the $i$-th anchor point in frame $t$, we select its $K$ nearest neighbors from the point cloud in frame $t+1$ and proceed to learn its local feature representation.

$$Q^{(t)} = \text{Ball Query}(D^{(t)}, P^{(t+1)}), \tag{3}$$

Here, Ball Query refers to the algorithm used for locating the points within a radius to the query point. $D^{(t)}$ denotes the anchor points in frame $t$, while $P^{(t+1)}$ designates the point cloud associated with frame $t+1$. By extracting geometric features across frames in this manner, the dynamics of the video are implicitly encoded into the local features.

**Group-Aware Aggregation.** Considering each cross-frame neighborhood feature discretely results in multiple complex, intertwined feature motion trajectories, complicating the estimation of the anchor's movement direction. To address this, we aggregate the obtained cross-frame local areas to generate a synthetic target, thereby guiding the subsequent learning of motion as follows:

$$E^{(t)} = \frac{1}{K} \sum_{j=1}^{K} Q_j^{(t)}, \tag{4}$$

**Group-Aware Motion Learning.** This step primarily predicts the motion trajectories of various groups based on the anchor point coordinates $\mathcal{D} \in \mathbb{R}^{M \times T \times 3}$ and the synthetic targets $E \in \mathbb{R}^{M \times T \times 3}$ obtained in the previous phase. This step involves no complex operations and merely requires index-wise subtraction operation. The process can be represented as follows:

$$\mathcal{X}^{(t)} = E^{(t)} - \mathcal{D}^{(t)}, \tag{5}$$

### 3.2.2 SINGLE-STEP MOTION ENCODER

Given an input point cloud video and the learned motion, the Single-Step Motion Encoder first utilizes the motion learned by the Motion Imitator to simulate dynamic virtual frames for each frame of the video. The encoder then focuses on the point cloud of the current frame, extracting local geometric information from the synthesized frames. The interaction between the current frame and the virtual frames inherently captures the dynamic information of the video. The features extracted through these queries are then fed into a PointNet-like network for further feature extraction. Through this single-step encoder, the network achieves a natural spatiotemporal understanding of the video, effectively bypassing the need for complex recurrent and dense query operations.

**Virtual Frame Synthesization.** This process is critical in embedding dynamic cues within point cloud videos. By employing the Motion Imitator, which is adept at capturing motion patterns from the input sequences, PvNeXt is able to generate virtual frames that embody the temporal evolution of the scene. The virtual frames are synthesized by adding the anchor-aware motion to the anchor-based groups. This synthesization process is mainly achieved as follow:

$$H'^{(t)} = \mathcal{X}^{(t)} + H^{(t)}, \tag{6}$$

Here, $H^{(t)}$ represents the anchor-based groups, obtained by searching for several neighbors within the point clouds. Frames within these groups are synthesized by interpolating motion trajectories in a group-wise manner, ensuring that the generated frames exhibit coherent and realistic motion characteristics. This approach not only enriches the raw frames with dynamic context but also amplifies the efficacy of the subsequent feature extraction process.

**One-Step Query Operation.** The one-step query operation is designed to streamline the extraction of spatio-temporal features by consolidating the querying process into a single step. Distinct from

iterative recurrent mechanisms or dense query strategies, this method executes the query operation only once to directly integrate interactions between current and virtual frames.

This approach not only efficiently captures both local geometries and dynamic information in a coherent framework but also significantly reduces computational overhead and enhances the efficiency of the processing pipeline. The integrity of motion and spatial details is preserved throughout this process. Subsequently, the features extracted via this singular query operation are refined further using a network akin to PointNet for enhanced feature delineation.

## 4 EXPERIMENTS

We assess the performance of our method for point cloud video recognition on various datasets, including MSR-Action3D (Li et al., 2010), and more challenging NTU-RGBD (Shahroudy et al., 2016). Furthermore, we provide ablative analyses on our core algorithm design.

### 4.1 EXPERIMENTAL SETUP

**Datasets.** We conduct experiments on two widely used datasets, MSR-Action3D and NTU-RGBD:

- **MSR-Action3D** comprises 567 depth videos captured via Kinect v1, spanning 20 distinct action categories and totaling approximately 23,000 frames. Each video contains an average of 40 frames. Following established protocols in recent studies (Fan et al., 2023), we partition the dataset into 270 training videos and 297 testing videos.

- **NTU-RGBD** encompasses 56,880 videos across 60 fine-grained action categories, providing a substantial variability in frame count per video, ranging from 30 to 300. Recorded using Kinect v2 and involving three cameras and 40 subjects, the dataset adopts a cross-subject evaluation methodology. In this method, subjects are bifurcated into two groups of 20 for training and testing purposes, respectively, resulting in 40,320 training videos and 16,560 test videos.

**Training Details.** Our model, is trained end-to-end, following distinct protocols as detailed below:

- **MSR-Action3D:** During training, 4/8/12/16/24 frames are densely sampled and 2048 points are selected in each frame. Notably, the frame step for 4-frame setting is defulat to 2, while 1 for other settings. The model is trained for 50 epochs with a batch size of 64 on single Geforce RTX 3090 GPU. We use the SGD optimizer and the initial learning rate is set to 0.01 with cosine decay.

- **NTU-RGBD:** Consistent with prior research (Fan et al., 2021a; 2023), the model processes 24 frames at each training step, each containing 2048 points, with a temporal step of 2. This dataset's training extends over 15 epochs with a batch size of 24, utilizing a single Geforce RTX 3090 GPU. The SGD optimizer is employed, setting the initial learning rate at 0.01 with a cosine decay.

**Baselines.** We compare our method with the state-of-the-art PSTNet++ and PST-Transformer.

- **PSTNet++.** Utilizing a temporal window, the PSTNet++ observes a limited set of frames for each localized region, thereby preserving the spatio-temporal structure. Nevertheless, this approach entails densely querying temporal neighbors, which substantially increases computation.

- **PST-Transformer**. The PST-Transformer decouples the spatio-temporal structure to reduce the impact of the spatial irregularity on the temporal modeling, adaptively searching related or similar points for raw point cloud video modeling. Though effective on modeling the 4D sequences, it suffers from the temporal looping and dense querying process.

### 4.2 EXPERIMENTS ON MSR-ACTION3D

**Quantitative Comparison.** The comparative performance of various methodologies in terms of point cloud video classification is systematically listed in Tab. 1, Flops and Parameters are achieved under the 16-frame setting. The results unequivocally demonstrate that our proposed method exhibits superior performance, registering an exemplary state-of-the-art accuracy of **94.77%** in 24-frame setting. This outstanding performance is notably achieved through straightforward network architectural modifications paired with our unique motion extraction approach, eschewing the need for intricate training scheme alterations or complex feature extraction. In direct comparison with the sota method PST-Transformer (Fan et al., 2023) with sophisticated 4D convolution operator, our

Table 1: Classification results on the MSR-Action3D dataset, accuracy(%, ↑) is reported.

| Methods | Reference | 8-frame | 12-frame | 16-frame | 24-frame | Flops(G) | Params(M) |
|---|---|---|---|---|---|---|---|
| *Supervised Learning Only* | | | | | | | |
| MeteorNet (Liu et al., 2019) | ICCV2019 | 81.14 | 86.53 | 88.21 | 88.50 | 1.7 | 17.6 |
| PSTNet (Fan et al., 2022) | ICLR2021 | 83.50 | 87.88 | 89.90 | 91.20 | 29.06 | 8.26 |
| P4Transformer (Fan et al., 2021a) | CVPR2021 | 83.17 | 87.54 | 89.56 | 90.94 | 32.55 | 44.14 |
| Kinet (Zhong et al., 2022) | CVPR2022 | 83.84 | 88.53 | 91.92 | 93.27 | 10.35 | 3.20 |
| PPTr (Wen et al., 2022) | ECCV2022 | 84.02 | 89.89 | 90.31 | 92.33 | - | - |
| PSTNet++ (Fan et al., 2021b) | TPAMI2022 | 83.50 | 88.15 | 90.24 | 92.68 | 30.21 | 8.43 |
| LeaF (Liu et al., 2023) | ICCV2023 | 84.50 | - | 91.50 | 93.84 | - | - |
| PST-Transformer (Fan et al., 2023) | TPAMI2023 | 83.97 | 88.15 | 91.98 | 93.73 | 32.56 | 44.13 |
| 3DInAction (Ben-Shabat et al., 2024) | CVPR2024 | 86.20 | 88.22 | 90.57 | 92.23 | - | 10.90 |
| PvNeXt (Ours) | - | **88.88** | **89.89** | **93.93** | **94.77** | **0.55** | **0.72** |
| *with Self-Supervised Representation Learning* | | | | | | | |
| C2P (Zhang et al., 2023) | CVPR2023 | 87.16 | - | 91.89 | 94.76 | - | - |
| PointCMP (Shen et al., 2023b) | CVPR2023 | 89.56 | 91.58 | 92.26 | 93.27 | - | - |
| PointCPSC (Sheng et al., 2023) | ICCV2023 | 88.89 | 90.24 | 92.26 | 92.68 | - | - |
| MaST-Pre (Shen et al., 2023a) | ICCV2023 | - | - | - | 94.08 | - | - |

Table 2: Comparisons on computational overhead between our method and other supervised learning methods on MSR-Action3D (16-frame) benchmark.

| Method | Reference | Memory (G) | Train speed (samples/s) | Infer speed (samples/s) | Accuracy (%) |
|---|---|---|---|---|---|
| PSTNet | ICLR2021 | 16.01 | 35.50 | 82.62 | 89.90 |
| P4Transformer | CVPR2021 | 9.97 | 43.02 | 99.03 | 89.56 |
| PSTNet++ | TPAMI2022 | 18.49 | 30.87 | 75.65 | 90.24 |
| PST-Transformer | TPAMI2023 | 11.17 | 30.48 | 75.33 | 91.98 |
| Ours (Batch Size=64) | - | 6.68 | 800.35 | 1735.45 | 93.93 |

method exhibits a marked enhancement of 1.04% accuracy lift. Moreover, APCT consistently attains high accuracy results across settings of different frames: 88.88%, 89.89%, and 93.93% for 8 / 12 / 16 frames, respectively. These metrics are either at the pinnacle or are approaching the current best results in each respective subcategory.

**Efficiency Comparison.** In order to demonstrate the efficacy of our proposed methodology, comprehensive comparative analyses were undertaken, the results of which are delineated in Tab. 2 and Fig. 4a. These analyses encompass evaluations of parameters, alongside training and inference speed. Consistent with experimental setups detailed in prior studies (Fan et al., 2022; 2021b;a; 2023), baseline models were trained with a batch size of 16, whereas our model employed 64.

Our methodology demonstrably surpasses preceding strategies in terms of both speed and accuracy. Specifically, when compared to the state-of-the-art method PST-Transformer (Fan et al., 2023), our model registered a **26**x speedup in training and a **23**x increase in inference speed. Concurrently, it exhibited an improvement in accuracy by 1.95% accuracy. With respect to the parameter count, our approach take merely **0.72**M parameters, presenting a substantial reduction from current methods. It utilizes only 1.6% (**0.72M** *v.s.* 44.13M) of the parameters necessitated by the PST-Transformer.

**Memory Analysis.** The proposed PvNeXt architecture not only achieves superior performance in terms of accuracy but also demonstrates an optimal accuracy versus memory consumption trade-off. This is illustrated in Fig. 4b, where PvNeXt shows a substantial reduction in GPU memory usage. Specifically, PvNeXt reduces memory consumption by **2.77**× compared to PSTNet++ and by **1.67**× relative to PST-Transformer. This improvement underscores PvNeXt's efficiency in utilizing computational resources while enhancing model performance.

## 4.3 EXPERIMENTS ON NTU-RGBD

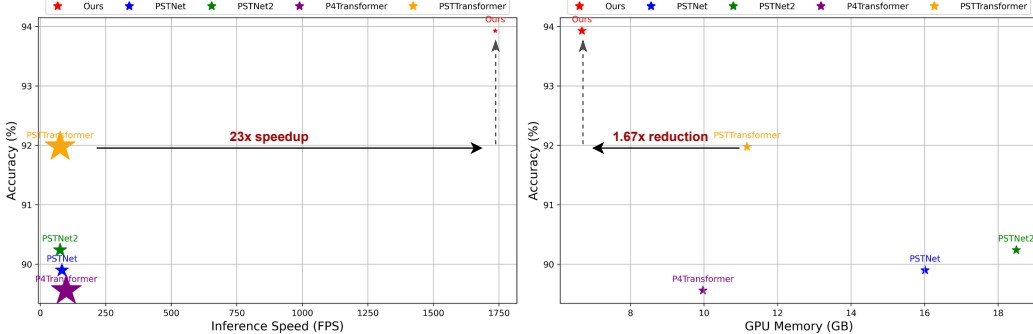

Figure 4: Comparisons between PvNeXt and other baselines on MSR-Action3D (16-frame). The size of the pentagram in (a) denotes the parameters, the larger shape denotes higher parameters.

Tab. 3 summarizes the comparison results on NTU-RGBD dataset, showing our algorithm works well on real-world challenging point cloud videos. In particular, our algorithm achieves impressive result of 89.2% accuracy. It is noteworthy that our method is remarkably simple and lightweight. It achieves results compared to those method of sophisticated feature extractors (*e.g.,* PSTNet (Fan et al., 2022), PST-Transformer (Fan et al., 2023) ) through mere extraction of motion and straightforward feature encoding. This clearly demonstrates the effectiveness of our algorithm, which learns advantageous information to assist in challenging video understanding. Besides, we experimented with integrating core components of our method with

Table 3: Classification results on NTU-RGBD under cross-subject setting, Action recognition accuracy($\%,\uparrow$) is reported. $^\dagger$ denotes semi-supervised on $50\%$ annotated data.

| Method | Reference | Accuracy |
|---|---|---|
| *Supervised Learning Only* | | |
| 3DV-Motion (Wang et al., 2020) | CVPR2020 | 84.5 |
| 3DV-PointNet++ (Wang et al., 2020) | CVPR2020 | 88.8 |
| PSTNet (Fan et al., 2022) | ICLR2021 | 90.5 |
| P4Transformer (Fan et al., 2021a) | CVPR2021 | 90.2 |
| PSTNet++ (Fan et al., 2021b) | TPAMI2022 | 91.4 |
| PvNeXt (Ours) | - | 89.2 |
| PST-Transformer (Fan et al., 2023) | TPAMI2023 | 91.0 |
| PST-Transformer + Ours | - | 91.4 |
| *with Self-Supervised Representation Learning* | | |
| PointCMP (Shen et al., 2023b) | CVPR2023 | 88.5 |
| PointCPSC$^\dagger$ (Sheng et al., 2023) | ICCV2023 | 88.0 |
| MaST-Pre$^\dagger$ (Shen et al., 2023a) | ICCV2023 | 90.8 |

the advanced PST-Transformer network. Specifically, we introduced an additional one-step query branch to the existing PST-Transformer structure to assist in extracting temporal features. This branch incorporates a Motion Imitator and a Single-Step Motion Encoder, designed to enhance the model's ability to handle dynamic sequences effectively. Equipped with our methodology, the PST-Transformer's accuracy arises from 91.0 to 91.4, achieving state-of-the-art performance.

In line with some self-supervised representation methods specifically designed for 4D sequences, our method distinctly surpasses them, delivering $0.7\%$ and $1.2\%$ accuracy increase for PointCMP (Shen et al., 2023b) and PointCPSC (Sheng et al., 2023), respectively.

## 4.4 ABLATION STUDIES

For in-depth analysis, we conduct ablative studies using 16-frame classification on MSR-Action3D.

**Motion Imitation.** We first investigate the effect of the motion imitation process, the results are listed in Tab. 4. Owe to this ingenious design, the extracted motion is gradually updated, aligned with and adaptive to the real dynamics of the video, making PvNeXt a compact model. To fully demonstrate the effectiveness, we study a variant, PvNeXt with Motion Imitator, where we only preserve the feature encoding process. As shown in table, a clear performance drop is observed, *i.e.,* acc: $93.93\% \rightarrow 86.86\%$, revealing the efficacy of our motion imitator.

**Effect of movement trajectories.** We investigate the impact of the virtual motion order of the future frames via learning motions. Specially, we reverse the movement trajectories by adjusting changes to the direction of motion, the results are listed in Tab. 5. Despite the flexible motion extraction facilitated by motion imitators, recklessly reversing the trajectories of characters movements poses challenges to video understanding, the accuracy drops from $93.93\%$ to $92.25\%$. This, in turn, reaffirms the efficacy of motion imitators in accurately depicting video dynamics.

Table 4: Effect of Motion Imitator.

| Encoder | Imitator | Accuracy |
|---------|----------|----------|
| ✓ | ✗ | 86.86 |
| ✓ | ✓ | 93.93 |

Table 5: Effect of movement trajectories.

| **PvNeXt** | Motion Objective | Accuracy |
|--------|------------------|----------|
| Reverse | $x - \Delta x$ | 92.25 |
| Forward | $x + \Delta x$ | 93.93 |

Table 6: Effect of neighbor number.

| Number | Accuracy |
|--------|----------|
| 1 | 93.27 |
| 3 | 93.93 |
| 5 | 93.60 |
| 7 | 92.93 |

Table 7: Comparison results with advanced architecture processing more frames.

| Method | Backbone | Frame | Acc(%) |
|--------|----------|-------|--------|
| MAMBA4D [Arxiv 2024] | Mamba | 32 | 93.38 |
| PvNeXt (Ours) | CNN | 16 / 24 | 93.93 / 94.77 |

Table 8: Results on extreme short frames (4-frame).

| Method | Acc(%) |
|--------|--------|
| PST-Transformer (Fan et al., 2023) | 81.14 |
| Ours | 75.75 |
| Ours + dense querying | 80.47 |

**Effect of neighbor number .** We investigate the impact of neighbor numbers of the Motion Imitator. As illustrated in Tab. 6, we vary the number of neighbors and analyze the results. The results show that the performance is 92.93% when the number is 7, which is 1.00% lower than the performance achieved using 3 neighbors. This observation can be attributed to the fact that too many neighbors can negatively impact the overall motion learning, therefore leading to a decrease in performance.

**Comparative analysis with advanced architectures handling extended frame sequences.** To mitigate the quadratic complexity inherent in transformer-based models, MAMBA4D (Liu et al., 2024) introduces an innovative 4D backbone specifically designed for point cloud video understanding, leveraging State Space Models (SSMs) to capture long-range dependencies with linear complexity. While this architecture excels at processing a greater number of frames, achieving a notable accuracy of 93.38% as seen in Tab. 7, our proposed method surpasses these results with 93.93% and 94.77% accuracy for 16 and 24 frames, respectively. The more advanced architecture of MAMBA4D still cannot avoid the multiple iterative loops and dense query operations in point cloud motion analysis, while our method significantly reduces this overhead. By employing such one-step query operation, PvNeXt efficiently capture motion dynamics, demonstrating superior computational efficiency and effectiveness.

**Analysis of performance on extreme short sequences.** Our proposed modules, particularly designed for efficient motion capture and analysis in point cloud videos, do indeed face challenges when applied to very short sequences, such as those comprising only four frames. The limited temporal extent in these scenarios constrains the model's ability to capture comprehensive motion dynamics, which is essential for accurate analysis. However, they perform exceptionally well with slightly longer sequences, such as 8-frame sequences, where they can more effectively utilize the available temporal information. To enhance our model's performance on very short sequences, we have incorporated our method with a dense querying strategy, as the 4-frame results in MSR-Action3D shown in the Table. 8. The results indicate that when applying this strategy, our model also achieves commendable performance of 80.47 in 4-frame settings. This improvement demonstrates our method's adaptability and potential for handling diverse sequence lengths effectively.

## 5 CONCLUSION

In this work, we introduce a new algorithm, tailored for efficient point cloud video analysis. Our algorithm incorporates an personalized one-shot query strategy, facilitating the efficient capture of motion, thereby enabling the spatio-temporal modeling for point cloud videos. Experimental evaluations on comprehensive benchmarks manifest its superiority.

*Limitations and applications.* While our method is efficient due to the one-shot query strategy, it may compromise the geometric detail captured compared to dense querying methods. In future work, we intend to delve deeper into this aspect. Efficient point cloud video perception algorithms have immense potential in various real-world applications, especially where dynamic 3D environmental understanding is crucial, such as in robotics and AR/VR.

**Acknowledgments** This work was financially supported by the National Natural Science Foundation of China (No. 62101032), the Young Elite Scientist Sponsorship Program of China Association for Science and Technology (No. YESS20220448), and the Young Elite Scientist Sponsorship Program of Beijing Association for Science and Technology (No. BYESS2022167).

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

# A APPENDIX

The supplementary material herein extends the discussion and analysis presented in the primary manuscript. It is structured as follows:

**Detailed architectures.** (§ A.1) : This section delves into the architecture details of our methodology across different datasets, offering a comprehensive understanding of our method.

**Comparison of our method with other modality results on NTU-RGBD dataset.** (§ A.2) : This section introduces the comparison analysis of our method with those of other modality on NTU-RGBD dataset.

**Evaluation of the learning performance of the motion imitator.** (§ A.3): This section introduces the analysis of the learning performance of the motion imitator.

**Performance in occlusion scenarios.** (§ A.4): This section introduces the analysis of the performance in occlusion scenarios.

**Results on HOI4D dataset.** (§ A.5): This section introduces the performance of our method on longer egocentric point cloud video dataset.

**Discussion about recognition tasks and dense action segmentation tasks.** (§ A.6): This section discuss the difference between recognition tasks and action segmentation tasks in point cloud videos.

## A.1 DETAILED ARCHITECTURES

At each learning stage, our method first generates the synthetic point clouds via the Motion Imitator, then it leverages the Single-Step Motion Encoder, which samples a subset of points from the virtual frames, to endow each sampled point with local geometric information and temporal motion. Through the accumulation of multiple stages, it progressively furnishes a smaller number of points, each enriched with an extended contextual awareness. The architecture of our method across different datasets is shown in Tab. 9.

Table 9: Architecture of PvNeXt across different datasets

| Dataset | Stage | MLPS | Nsamples | Spatial stride | Radius |
|---------|-------|------|----------|----------------|--------|
| MSR-Action3D | S1 | [64] | 48 | 32 | 0.2 |
| | S2 | [128], [128,256] | 32 | 8 | 0.4 |
| | S3 | [512], [512,1024] | 8 | 2 | 0.4 |
| NTU-RGBD | S1 | [64] | 32 | 8 | 0.1 |
| | S2 | [128], [128,256] | 48 | 8 | 0.2 |
| | S3 | [128], [128,256] | 16 | 1 | 0.4 |
| | S4 | [128], [128,256] | 24 | 1 | 0.4 |
| | S5 | [512], [512,1024] | 32 | 4 | 0.8 |

## A.2 COMPARISON OF OUR METHOD WITH OTHER MODALITY RESULTS ON NTU-RGBD DATASET.

### A.2.1 MOTIVATION OF DATA CONVERSION.

Our method involves converting RGBD data into point clouds. This step is common in the field, primarily because it allows researchers to exploit the rich set of tools and algorithms developed for 3D point cloud processing. While this conversion introduces an additional step, it also opens up opportunities to leverage point cloud-specific methodologies that might not be directly applicable to RGBD images, particularly those that exploit the inherent 3D structure of the data.

The conversion from RGBD to point clouds might not always lead to enhanced efficiency in processing. In some cases, direct RGBD image processing can be faster due to optimized 2D convolutional operations available in current deep learning frameworks. However, the conversion to point clouds is often justified by the ability to handle and interpret 3D data more naturally and accurately, especially for tasksrequiring precise depth estimations and spatial relationships.

With the advancement of sensors such as LiDAR, a substantial influx of point cloud data is expected in the near future. Therefore, it is essential to explore direct processing of point cloud sequences for recognition tasks, which may be a new direction in the field.

### A.2.2 DATA AND SETUP.

The NTU-RGBD dataset encompasses 3D skeletons (5.8GB), masked depth maps (83GB), and RGB videos (136GB). Our point cloud data is derived solely by extracting depth positions from masked depth maps and converting them, not by combining RGB videos and depth maps. Consequently, the derived point cloud data inherently contains less information than RGB videos due to reduced richness of source data.

### A.2.3 COMPARISON ANALYSIS.

During our experiments, the input point cloud for our proposed method solely **derives from depth maps**. We do not utilize any auxiliary information such as skeletons or RGB images. As seen in Tab. 13, we provide detailed comparison between methods processing various modal data, including *3D Skeleton, depth maps, RGB videos and point clouds*. Compared to methods that directly process depth maps, point cloud based approachs achieve superior performance. Since both derive from the same data source, this validates the potential of point cloud based methods in handling dynamic sequences.

Although effective, point cloud methods fall short when compared to the latest methods that directly process RGB videos. Our analysis of RGB video-based methods reveals several reasons for their higher performance. Recent algorithms for processing RGB video often employ self-supervised learning techniques, benefiting from the extensive availability of large-scale RGB video datasets. Recent RGB video algorithms typically utilize 32 or more frames, leveraging more temporal information than point cloud methods, which only use up to 24 frames. Direct comparisons with these RGB video methods are unreliable due to the significant computational overhead entailed by self-supervised techniques and more utilized frames. In contrast, our approach is designed to be highly efficient and lightweight, avoiding the substantial computational costs associated with these methods.

### A.3 EVALUATION OF THE LEARNING PERFORMANCE OF THE MOTION IMITATOR.

We further evaluate the learning performance of the motion imitator. Specifically, we compute the chamfer distance between the predicted point cloud and the ground truth (GT) point cloud. We compare the distance between the predicted point cloud (generated by our method) and the ground truth (GT) point cloud with the distance between the predicted point cloud (generated by nearest-neighbor sampling) and the GT point cloud. Nearest-neighbor sampling strategy, the paradigm commonly adopted in prior approaches (Fan et al., 2022; 2021b;a; 2023), serves as the baseline for this comparison.

As shown in the Tab. 10, the chamfer distance produced by our method is significantly lower, than that of the baseline, indicating that the motion imitator learns correlations more effectively. This reduction indicates that our motion imitator effectively predicts point cloud displacements and aligns them closely with ground truth motion. The superior learning of motion dynamics facilitates better temporal consistency across frames, which improves downstream performance in tasks like action recognition.

Table 10: The learning performance of the motion imitator.

| Method | Chamfer Distance($\downarrow$) |
|---|---|
| Nearest-neighbor sampling | 5.23e-3 |
| Ours | **1.35e-3** |

### A.4 Performance in occlusion scenarios.

We further evaluate the performance of our approach in occlusion scenarios. We adopted a simulation-based approach to validate the effectiveness of our method. Specifically, we simulated the occlusion effect encountered by LiDAR sensors, which often leads to partial data loss. Inspired by the Drop-Local corruption operation from the ModelNet-C (Ren et al., 2022), we applied a similar strategy to the MSR-Action3D dataset. This approach artificially introduced local point cloud occlusions to simulate real-world challenges and constructed a simplified occlusion benchmark for evaluating the performance of our approach in such scenarios.

As shown in Tab. 11, our approach achieves an accuracy of 88.22% under occlusions, outperforming the advanced PST-Transformer by 1.69%. This result highlights the ability of our method to effectively capture spatial-temporal dynamics even in the presence of partial data loss. These results demonstrate that the Motion Imitator and Single-Step Motion Encoder components of our framework can effectively model motion dynamics, even when some data points are missing. This robustness suggests that our method remains effective in scenarios involving occlusions.

Table 11: Performance in occlusion scenarios.

| Method | Accuracy | Accuracy$_{occ}$ |
|---|---|---|
| PST-Transformer | 91.98 | 86.53 |
| Ours | **93.93** | **88.22** |

### A.5 Results on HOI4D dataset.

We further evaluate the performance of our approach on the additional HOI4D (Liu et al., 2022) dataset,a large-scale 4D egocentric dataset that captures diverse, category-level human-object interactions. HOI4D comprises 2.4M RGB-D egocentric video frames over 4000 sequences, collected by 9 participants interacting with 800 unique object instances spanning 16 categories in 610 different indoor rooms. We evaluated our method on the dataset's action segmentation task, which includes 2971 training scenes and 892 test scenes. Each sequence contains 150 frames, with each frame represented by 2048 points. The task requires the model to predict an action label for each frame within a point cloud sequence.

As presented in the Tab. 12, our approach achieves an accuracy of 78.5%, outperforming the advanced method, PPTr, which achieves an accuracy of 77.4%. This improvement highlights the effectiveness of our model in capturing fine-grained motion dynamics, thereby demonstrating robust generalization capabilities in real-world scenarios. Furthermore, we observed consistent improvements in additional metrics, such as Edit and F1 scores, highlighting our model's superior ability to handle dense prediction tasks.

These improvements demonstrate our model's capability to better align temporal predictions with ground truth actions, reduce segmentation errors, and handle transitions smoothly between actions. These results underline the robustness of our framework in capturing fine-grained motion dynamics over long sequences.

### A.6 Discussion about recognition tasks and dense action segmentation tasks.

In point cloud videos, recognition tasks inherently exist information redundancy across point cloud frames, which our one-shot query strategy effectively handles. In contrast, dense prediction tasks, such as action segmentation, require a deeper understanding of motion information for each frame, presenting significantly different challenges.

Our method is primarily designed for point cloud video recognition tasks, offering limited improvements in dense prediction tasks. We believe this distinction between the two task types is an important and valuable topic for the community to explore further. We hope this discussion will encourage others to address these challenges and explore innovative solutions.

Table 12: Results on HOI4D Action segmentation dataset.

| Method | Length | Accuracy | Edit | F1@10 | F1@25 | F1@50 |
|---|---|---|---|---|---|---|
| P4Transformer | 150 | 71.2 | 73.1 | 73.8 | 69.2 | 58.2 |
| PPTr | 150 | 77.4 | 80.1 | 81.7 | 78.5 | 69.5 |
| Ours | 150 | **78.5** | 84.5 | 85.6 | 82.4 | 73.0 |

Table 13: Classification results on NTU-RGBD under cross-subject setting, Action recognition accuracy($\%, \uparrow$) is reported. $*$ denotes self-supervised representation learning. $\dagger$ denotes semi-supervised on $50\%$ annotated data.

| Method | Input | Accuracy($\%, \uparrow$) | Flops(G) | Params(M) |
|---|---|---|---|---|
| *3D Skeleton* | | | | |
| SkeleMotion (Caetano et al., 2019) | *Skeleton* | 69.6 | - | - |
| DGNN (Shi et al., 2019) | *Skeleton* | 89.9 | - | - |
| AGC-LSTM (Si et al., 2019) | *Skeleton* | 89.2 | - | - |
| Shift-GCN (Cheng et al., 2020) | *Skeleton* | 90.7 | - | - |
| ActCLR (Lin et al., 2023) | *Skeleton* | 88.2 | - | 2.5 |
| SkeAttnCLR (Hua et al., 2023) | *Skeleton* | 89.4 | - | - |
| Js-SaPR-GCN (Li et al., 2023) | *Skeleton* | 90.1 | 1.7 | 2.1 |
| CTR-GCN (Chen et al., 2021) | *Skeleton* | 92.4 | - | - |
| BlockGCN (Zhou et al., 2024) | *Skeleton* | 90.9 | - | 1.5 |
| *Depth maps* | | | | |
| DynamicMaps+CNN (Wang et al., 2018) | *Depth* | 87.1 | - | - |
| MVDI (Xiao et al., 2019) | *Depth* | 84.6 | - | - |
| PA-AWCNN (Yao et al., 2022) | *Depth* | 89.6 | - | - |
| ActionMAE (Woo et al., 2023) | *Depth* | 90.1 | - | - |
| *RGB videos* | | | | |
| Chained Multi-stream (Zolfaghari et al., 2017) | *RGB* | 80.8 | - | - |
| Glimpse Clouds (Baradel et al., 2018) | *RGB* | 86.6 | - | - |
| CNN+bi-LSTM(Debnath et al., 2021) | *RGB* | 87.2 | - | 9.4 |
| PA-AWCNN (Yao et al., 2022) | *RGB* | 90.4 | - | - |
| ViewCLR (Das & Ryoo, 2023) | *RGB* | 89.7 | - | - |
| Multi-View Learning (Shah et al., 2023) | *RGB* | 91.4 | - | - |
| MV2MAE (Shah et al., 2024) | *RGB* | 90.0 | - | - |
| *Point clouds* | | | | |
| 3DV-Motion (Wang et al., 2020) | *Point* | 84.5 | - | - |
| 3DV-PointNet++ (Wang et al., 2020) | *Point* | 88.8 | - | - |
| PSTNet (Fan et al., 2022) | *Point* | 90.5 | 19.6 | 8.5 |
| P4Transformer (Fan et al., 2021a) | *Point* | 90.2 | 48.6 | 65.2 |
| PSTNet++ (Fan et al., 2021b) | *Point* | 91.4 | - | - |
| PointCMP$^*$ (Shen et al., 2023b) | *Point* | 88.5 | - | - |
| PointCPSC$^{*\dagger}$ (Sheng et al., 2023) | *Point* | 88.0 | - | - |
| MaST-Pre$^{*\dagger}$ (Shen et al., 2023a) | *Point* | 90.8 | - | - |
| PvNeXt (Ours) | *Point* | 89.2 | 7.8 | 0.9 |
| PST-Transformer (Fan et al., 2023) | *Point* | 91.0 | 48.6 | 65.2 |
| PST-Transformer + Ours | *Point* | 91.4 | 58.4 | 77.8 |

