# OpenReview forum: "PvNeXt: Rethinking Network Design and Temporal Motion for Point Cloud Video Recognition"
_ICLR.cc/2025/Conference — ICLR 2025 Poster_

### Official Review · Reviewer_eQ8c · 2024-11-01

**Soundness:** 3
**Presentation:** 3
**Contribution:** 3
**Rating:** 6
**Confidence:** 2

**Summary:**

Considering traditional method capture temporal features effectively but at a high computational cost. The paper addresses the computational challenges in point cloud video perception by introducing PvNeXt, an efficient framework for 3D video recognition. PvNeXt introduces a more efficient approach through two main modules: the Motion Imitator models temporal dynamics by generating virtual motion for each frame and the Single-Step Motion Encoder captures motion cues and temporal dynamics in a single step.

**Strengths:**

1. PvNeXt reduces computational redundancy by replacing iterative, dense query operations with a one-shot query mechanism, enabling faster inference and lower training costs.
2. In experiment, the PvNeXt achieve SOTA especially in flops and parameters.
3. In experiment, the paper provide comprehensive ablation studies and discussion.

**Weaknesses:**

The conclusion is too short, lack of analysis of network limitations and discussion of application scenarios since the network is lightweight. Are there particular types of limitations you think should be discussed? Or specific application areas where the lightweight nature could be especially impactful?

**Questions:**

Can the method proposed in the paper still effectively solve the problem of longer time-series data or very fast high-speed motion?

---

> ### Author Response · Authors · 2024-11-23
> **Response to Reviewer eQ8c**
>
> We sincerely appreciate your detailed and insightful reviews. We have provided detailed responses to your comment and updated the relevant content in the revised manuscript, hoping our response can address your concerns.
>
> >**Q1: The conclusion is too short, lack of analysis of network limitations and discussion of application scenarios.**
>
> Thanks for your advice! It is very necessary to discuss the limitations and potential application scenarios of our model.
>
> 1. Limitations: Our one-shot query strategy, designed for high efficiency, sometimes compromises the depth of geometric detail captured compared to dense querying methods. This trade-off, while beneficial for processing speed and resource utilization, can affect the model's ability to handle complex geometric structures in point cloud data.
>
> 2. Application scenarios: Efficient point cloud video perception algorithms have immense potential in various real-world applications, especially where dynamic 3D environmental understanding is crucial. Here's an analysis of their potential applications:
>     * Robotics: Enable real-time navigation, precise object manipulation, and coordinated swarm operations.
>     * AR/VR: Enable immersive experiences through precise motion tracking and seamless integration of virtual content with real-world environments.
>     * Autonomous Vehicles: Facilitate real-time mapping, dynamic object detection, and predictive trajectory modeling to enhance safety and decision-making in complex traffic scenarios.
>
> We have updated conclusion as suggested. We hope this extended discussion provides a more comprehensive perspective and addresses your concerns.  Thank you for your constructive feedback!
>
> >**Q2: Can the method proposed in the paper still effectively solve the problem of longer time-series data or very fast high-speed motion ?**
>
> Thanks for your insightful question!  It is very necessary to discuss the performance of our model on longer time-series data or very fast high-speed motion.
>
> As you suggested, we further evaluate the performance of our approach on the additional **HOI4D**[1] dataset, a large-scale 4D egocentric dataset that captures diverse and category-level human-object interactions, it comprises 2.4M RGB-D egocentric video frames over 4000 sequences. We evaluated our method on the dataset’s action segmentation task, which includes 2971 training scenes and 892 test scenes. Each sequence contains **150** frames, with each frame represented by 2048 points.  This dataset is particularly suitable for assessing the model’s ability to handle extended time-series data. The task requires the model to predict an action label for each frame within a point cloud sequence.
>
> As presented in the table below, our approach achieves an accuracy of 78.5%, outperforming the advanced method, PPTr, which achieves an accuracy of 77.4%. Additionally, our approach demonstrates superior performance in metrics like Edit Distance and F1 scores, indicating its robustness in handling longer time-series data (150 frames). All these results highlight the model’s ability to effectively capture temporal dependencies and motion dynamics over longer sequences.
>
> While the HOI4D dataset includes diverse motion dynamics, it does not explicitly focus on scenarios with very fast high-speed motion. To address this, future evaluations could involve synthetic datasets or benchmarks specifically designed to simulate rapid temporal changes, such as high-frame-rate point cloud sequences or scenarios with significant motion blur.
>
> We appreciate your suggestion and recognize the importance of addressing these challenges. Future work will focus on expanding our evaluations to include high-speed motion datasets and further refining the model for such scenarios. Thank you for your valuable feedback!
>
> | Method       | Length  | Accuracy | Edit| F1@10 | F1@25 | F1@50 |
> |--------------|---------| ---------| --- |------ |------ |------ |
> |P4Transformer | 150 | 71.2 | 73.1 | 73.8 | 69.2 | 58.2 |
> | PPTr         | 150 | 77.4 | 80.1 | 81.7 | 78.5 | 69.5 |
> | Ours         | 150 | **78.5**	| 84.5 | 85.6 |	82.4 | 73.0 |
>
> Reference:
>
> [1] HOI4D: A 4D Egocentric Dataset for Category-Level Human-Object Interaction CVPR 2022.

---

### Official Review · Reviewer_Gk9h · 2024-11-01

**Soundness:** 3
**Presentation:** 3
**Contribution:** 3
**Rating:** 8
**Confidence:** 5

**Summary:**

The article proposes a point cloud video classification framework, rethinks the core issues of framework design and provides reasonable solutions. The article uses two key modules, Motion Imitator and the Single-Step Motion Encoder, to significantly accelerate the classification efficiency of point cloud videos. The authors also verified the effectiveness of this scheme on two human datasets.

**Strengths:**

1. The problem of understanding point cloud videos is very important. This article rethinks the key challenges and network design in this field, providing inspiration for progress in this direction.
2. The article points out two core challenges of current networks in terms of computational overhead, the need to traverse all frames and the overlap of local point cloud patches. This observation is reasonable.
3. The article found that effective feature extraction can be achieved by only performing one-step queries between local regions between adjacent frames, which is an interesting discovery.
4. This experimental finding also shows that we need more new 4D tasks, in addition to classification, there are also segmentation, prediction, etc.
5. The previous method proposed to reduce the range of the receptive field in the spatial dimension. This article proposes to reduce the receptive field in the temporal dimension and achieve good performance.

**Weaknesses:**

1. Motion imitator implicitly performs point tracking to learn the adjacent frame correlation of dense points. The single step motion encoder learns motion information to encode features based on the learned correlations. If I understand correctly, how does the author measure the correlation learned by the motion imitator? Because this correlation is easy to learn on the human body, it becomes significantly more difficult when other objects or backgrounds are included.
2. The author can calculate calculate the distance between the predicted human point cloud and the GT point cloud to evaluate the ability of the motion imitator. I think this link is easy to learn on the human dataset, and can even be calculated manually by some methods, but when it includes occlusion problems, scene backgrounds, and interactive objects, can it still be learned? This conclusion is very important for the design of future 4D networks because it determines whether we need to perform point tracking explicitly or implicitly.
3. Because NTU-RGBD contains background scene information, its results are not better than the baeline of PST-Transformer, which may also confirm the limitations of the method when the motion imitator is difficult to learn.
4. This paradigm of the article is only experimented on human point cloud recognition. Can this method be extended to point cloud video action segmentation tasks such as HOI4D action segmentation task, because these tasks require predicting a label for each frame and are more relevant to robotics or AR/VR applications. I am very curious whether the article method is effective in this dense prediction task.

**Questions:**

I think the article proposes a very interesting perspective to redesign the network architecture of point cloud videos, and this research is inspiring. However, some conclusions are still unclear, such as how to evaluate the learning performance of the motion imitator, whether point tracking is still easy to learn in other scenarios, and how this method performs on dense prediction tasks like HOI4D action segmentation task. The author can try to explain these key concerns, and I am very willing to improve the score.

---

> ### Author Response · Authors · 2024-11-23
> **Response to Reviewer Gk9h (Part 1)**
>
> We sincerely appreciate your detailed and insightful reviews. We have provided detailed responses to your comment and updated the relevant content in the revised manuscript, hoping our response can address your concerns.
>
> >**Q1: Evaluation of the learning performance of the motion imitator.**
>
> Thank you for your advice! It is very necessary to clarify the evaluation of the learning performance of the motion imitator.
>
> To further evaluate the learning performance of the motion imitator, we compute the chamfer distance between the predicted point cloud and the ground truth (GT) point cloud. We compare the distance between the predicted point cloud (generated by our method) and the ground truth (GT) point cloud with the distance between the predicted point cloud (generated by nearest-neighbor sampling) and the GT point cloud. Nearest-neighbor sampling strategy, the paradigm commonly adopted in prior approaches[1,2,3,4], serves as the baseline for this comparison.
>
> As shown in the table below, the chamfer distance produced by our method is significantly lower, than that of the baseline, indicating that the motion imitator learns correlations more effectively. This reduction indicates that our motion imitator effectively predicts point cloud displacements and aligns them closely with ground truth motion. The superior learning of motion dynamics facilitates better temporal consistency across frames, which improves downstream performance in tasks like action recognition.
>
> We hope this explanation provides clarity on the evaluation of the motion imitator's learning performance. Thanks for your valuable feedback!
>
> | Method            | Chamfer Distance($\downarrow$) |
> |-------------------|----------|
> | Nearest-neighbor sampling   |  5.23e-3    |
> | Ours                        |  **1.35e-3**|
>
> Reference:
>
>
> [1] PSTNet: Point Spatio-Temporal Convolution on Point Cloud Sequences. ICLR 2021.
>
> [2] Point 4D Transformer Networks for Spatio-Temporal Modeling in Point Cloud Videos. CVPR 2021.
>
> [3] Deep Hierarchical Representation of Point Cloud Videos via Spatio-Temporal Decomposition. TPAMI 2022.
>
> [4] Point Spatio-Temporal Transformer Networks for Point Cloud Video Modeling. TPAMI 2023.
>
> >**Q2: Whether point tracking is still easy to learn in other scenarios, such as occlusion problems.**
>
> Thanks for your advice! Evaluating the performance of our model in occlusion scenarios is indeed crucial for understanding its practical applicability in real-world environments.
>
> As you suggested, we further evaluate the performance of our approach in occlusion scenarios. While we could not access a dedicated point cloud video dataset explicitly designed to include occlusion scenarios, we adopted a simulation-based approach to validate the effectiveness of our method.
> Specifically, we simulated the occlusion effect encountered by LiDAR sensors, which often leads to partial data loss. Inspired by the Drop-Local corruption operation from the ModelNet-C[5], we applied a similar strategy to the MSR-Action3D dataset. This approach artificially introduced local point cloud occlusions to simulate real-world challenges and constructed a simplified occlusion benchmark for evaluating the performance of our approach in such scenarios.
>
> As shown in the table below, our approach achieves an accuracy of 88.22% under occlusions, outperforming the advanced PST-Transformer by 1.69%. This result highlights the ability of our method to maintain robust point tracking and effectively capture spatial-temporal dynamics even in the presence of partial data loss. These results demonstrate that the Motion Imitator and Single-Step Motion Encoder components of our framework can effectively model motion dynamics, even when some data points are missing. This robustness suggests that point tracking with our method remains learnable and effective in scenarios involving occlusions.
>
> While the simulation provides valuable insights, real-world occlusions may introduce additional complexities, such as dynamic occlusion patterns or dense clutter. We see this as an exciting direction for future work, including experiments on more diverse datasets or real-world scenarios explicitly designed for occlusion handling.
>
> We hope this analysis addresses your concerns and showcases the robustness of our approach. Thank you for your valuable feedback!
>
> | Method            | Accuracy |Accuracy_occ |
> |-------------------|----------|-------------|
> | PST-Transformer   |  91.98    |    86.53     |
> | Ours              |  **93.93**    |    **88.22**     |
>
> Reference:
>
> [5] Benchmarking and Analyzing Point Cloud Classification under Corruptions. ICML 2022.

---

> > ### Author Response · Authors · 2024-11-23
> > **Response to Reviewer Gk9h (Part 2)**
> >
> > >**Q3: How this method performs on dense prediction tasks like HOI4D action segmentation task.**
> >
> > Thanks for your advice! It is very necessary to discuss the performance of our model on dense prediction tasks like HOI4D action segmentation task.
> >
> > As you suggested, we further evaluate the performance of our approach on the additional **HOI4D**[6] dataset,a large-scale 4D egocentric dataset that captures diverse, category-level human-object interactions. HOI4D comprises 2.4M RGB-D egocentric video frames over 4000 sequences, collected by 9 participants interacting with 800 unique object instances spanning 16 categories in 610 different indoor rooms. We evaluated our method on the dataset’s action segmentation task, which includes 2971 training scenes and 892 test scenes. Each sequence contains 150 frames, with each frame represented by 2048 points. The task requires the model to predict an action label for each frame within a point cloud sequence.
> >
> > As presented in the table below, our approach achieves an accuracy of 78.5%, outperforming the advanced method, PPTr, which achieves an accuracy of 77.4%. This 1.1% improvement highlights the effectiveness of our model in capturing fine-grained motion dynamics, thereby demonstrating robust generalization capabilities in real-world scenarios. Furthermore, we observed consistent improvements in additional metrics, such as Edit and F1 scores, highlighting our model’s superior ability to capture fine-grained motion dynamics and handle dense prediction tasks over long sequences.
> >
> > We hope this evaluation clarifies the performance of our method on dense prediction tasks and highlights its potential for real-world applications. Thank you for your valuable feedback!
> >
> > | Method       | Length  | Accuracy | Edit| F1@10 | F1@25 | F1@50 |
> > |--------------|---------| ---------| --- |------ |------ |------ |
> > |P4Transformer | 150 | 71.2 | 73.1 | 73.8 | 69.2 | 58.2 |
> > | PPTr         | 150 | 77.4 | 80.1 | 81.7 | 78.5 | 69.5 |
> > | Ours         | 150 | **78.5**	| 84.5 | 85.6 |	82.4 | 73.0 |
> >
> > Reference:
> >
> > [6] HOI4D: A 4D Egocentric Dataset for Category-Level Human-Object Interaction CVPR 2022.

---

> > > ### Comment · Reviewer_Gk9h · 2024-11-29
> > > **How does the one-shot query strategy perform frame-wise dense prediction?**
> > >
> > > I am not clear on how the one-shot query strategy is adapted for the dense prediction task on HOI4D, as this task is significantly different from classification tasks. In this case, do you need to retain the features of each frame for classification? If so, the computational overhead compared to traditional methods might not have a significant advantage. Could the authors explain the specific approach and the computational cost in this scenario?

---

> > > > ### Comment · Reviewer_Gk9h · 2024-12-01
> > > >
> > > > Has the author provided an answer to this question? This is crucial for my final rating.

---

> > > > > ### Author Response · Authors · 2024-12-01
> > > > >
> > > > > Thanks you for your insightful question! We acknowledge the differences between classification tasks and dense prediction tasks like action segmentation on HOI4D.
> > > > >
> > > > > Due to time constraints, we were unable to identify a direct method to adapt the one-shot query strategy for the dense prediction task. As an alternative, we leverage one of the core components of our method, the motion imitator, within the framework of PPTr. Specifically, we take the motion imitator to generate enhanced point clouds, which replace the original point clouds in the PPTr network, while keeping the rest of the network structure unchanged.
> > > > >
> > > > > This approach allow us to evaluate the effectiveness of the motion imitator in improving dense prediction performance. As shown in the table below, integrating the motion imitator improves the network's performance on the HOI4D action segmentation task, demonstrating the utility and generalizability of our method. Importantly, this enhancement is achieved without increasing the computational cost of the original network, maintaining efficiency while boosting performance.
> > > > >
> > > > > We hope this explanation addresses your concerns and illustrates how our method contributes to dense prediction tasks. Thank you again for your thoughtful feedback, and we look forward to further exploring the adaptation of the one-shot query strategy for dense prediction in future work.
> > > > >
> > > > > | Method       | Length  | Accuracy | Edit| F1@10 | F1@25 | F1@50 | FLOPs(G) | Params(M) |
> > > > > |--------------|---------| ---------| --- |------ |------ |------ | ------ |------ |
> > > > > | PPTr         | 150 | 77.4 | 80.1 | 81.7 | 78.5 | 69.5 | 622.2 | 128.1 |
> > > > > | Ours         | 150 | **78.5**	| 84.5 | 85.6 |	82.4 | 73.0 | 622.2 | 128.1 |

---

> > > > > > ### Comment · Reviewer_Gk9h · 2024-12-01
> > > > > >
> > > > > > Here are two follow-up questions:
> > > > > > 1. The motion imitator introduces additional computational overhead, so why do the FLOPs and Params remain unchanged?
> > > > > > 2. Why do predictions using the motion imitator perform better than the ground truth original point cloud? This result seems somewhat counterintuitive. Did the authors provide any explanation for this?

---

> > > > > > > ### Author Response · Authors · 2024-12-01
> > > > > > >
> > > > > > > We sincerely appreciate your detailed and insightful reviews. We hope our response can address your concerns.
> > > > > > > >**Q1: The motion imitator introduces additional computational overhead, so why do the FLOPs and Params remain unchanged?**
> > > > > > >
> > > > > > > Thanks for your insightful question! While it is true that the motion imitator introduces additional computational overhead, the increase is extremely small and negligible in practice. For reference, as demonstrated in our evaluation on the MSR-Action3D dataset, even when including the encoder's computations, the overall computational cost remains minimal. This small overhead does not significantly affect the reported FLOPs and Params values, allowing us to maintain efficiency comparable to the baseline models.
> > > > > > >
> > > > > > > >**Q2: Why do predictions using the motion imitator perform better than the ground truth original point cloud? This result seems somewhat counterintuitive. Did the authors provide any explanation for this?**
> > > > > > >
> > > > > > > Thanks for your insightful question! We agree that this result may seem counterintuitive at first glance. We own it to the motion imitator enhances performance by introducing a degree of data diversity into the training process. By generating slightly varied point clouds that retain meaningful motion cues, the motion imitator effectively enriches the data, reducing the risk of overfitting. This augmented diversity allows the model to generalize better, ultimately leading to improved performance compared to predictions based solely on the ground truth point cloud.
> > > > > > >
> > > > > > > We hope these explanations clarify the impact of the motion imitator. Thank you again for your thoughtful questions.

---

> > > > > > > > ### Comment · Reviewer_Gk9h · 2024-12-01
> > > > > > > >
> > > > > > > > Thank you for the author's response. However, this still does not address my concern. If the motion imitator introduces more uncertainty, it could lead to many incorrect labels. And when predicting the point cloud at time t, does the author only replace the point clouds after time t? This would clearly affect the accuracy of frame t. Although this experiment is just supplementary and does not impact the author's contribution to the classification task, I find the experimental results very confusing and believe that this could mislead the community.

---

> > > > > > > > > ### Author Response · Authors · 2024-12-02
> > > > > > > > >
> > > > > > > > > We sincerely appreciate your detailed and insightful reviews. We hope our response can address your concerns.
> > > > > > > > > >**Q1: If the motion imitator introduces more uncertainty, it could lead to many incorrect labels.**
> > > > > > > > >
> > > > > > > > > Thanks for pointing this out! Your concern about the potential for incorrect labels caused by increased uncertainty is indeed valid, and we appreciate the opportunity to clarify.
> > > > > > > > >
> > > > > > > > > We were also cautious about this issue and implemented a specific mitigation strategy to address it. To reduce the risk of incorrect labels, we assigned the time identifier of the subsequent frame (like https://github.com/hoi4d/HOI4D_ActionSeg/blob/main/models/AS_pptr_base.py/#L61) to the virtual frame generated by the motion imitator. This ensures that the temporal correspondence between frames is preserved, which helps mitigate the uncertainty introduced by the motion imitator.
> > > > > > > > >
> > > > > > > > > We hope this explanation alleviates your concerns and clarifies our approach to handling this issue. Thank you again for raising such an important point!
> > > > > > > > >
> > > > > > > > > >**Q2: And when predicting the point cloud at time t, does the author only replace the point clouds after time t?**
> > > > > > > > >
> > > > > > > > > Thanks for your question! To clarify, we do not predict and replace the point clouds for frames after time t. Instead, we get all the virtual frames in advance using the motion imitator. These virtual frames are then input into the network's encoder as a complete sequence, ensuring that the temporal relationships across the entire sequence are preserved and effectively utilized.
> > > > > > > > >
> > > > > > > > > Specifically, in the PPTr framework, our operation is performed on the outputs of the first encoder`self.tube_embedding()`(https://github.com/hoi4d/HOI4D_ActionSeg/blob/main/models/AS_pptr_base.py/#L49), rather than the initial input. By integrating the virtual frames at this stage, we enhance the representation learning ability without changing the fundamental structure of the network.
> > > > > > > > >
> > > > > > > > > We hope this explanation alleviates your concerns. Thank you again for raising such an important point!

---

> > > > > > > > > > ### Comment · Reviewer_Gk9h · 2024-12-03
> > > > > > > > > > **Some Suggestion**
> > > > > > > > > >
> > > > > > > > > > Thank you for the author's response.  I still do not understand why the motion imitator would work on dense prediction. Anyway, we can agree that the method does not bring computational savings in dense prediction tasks (in fact, there is an increase), and the improvement over previous methods is limited.
> > > > > > > > > >
> > > > > > > > > > I suggest the authors honestly discuss the differences between recognition tasks and action segmentation tasks, and limit their method to recognition tasks, without overclaiming results in action segmentation. Because the authors have already adequately demonstrated the contribution of their method in recognition tasks and have shown that there is a lot of information redundancy in recognition tasks, which can be addressed with a one-shot approach. Unlike recognition tasks, dense prediction tasks require the model to understand the motion information of each point cloud frame, and the issues and challenges are significantly different. I think this message is beneficial to the community because it inspires us to build more dense 4D tasks. I believe that in the in-depth discussions, it also made me think about the differences between recognition tasks and dense prediction tasks, which is a quite interesting problem.
> > > > > > > > > >
> > > > > > > > > > If the authors honestly discuss the challenges of the two types of problems and the limitations of their method in dense prediction (which I don't think is a significant weakness), I believe it will inspire more people to pay attention to the task definition of 4D point cloud videos. If that's the case, I would consider raising my score.

---

> > > > > > > > > > > ### Author Response · Authors · 2024-12-03
> > > > > > > > > > >
> > > > > > > > > > > Thanks for your thoughtful feedback and valuable suggestions! We greatly appreciate your insights, especially regarding the differences between recognition tasks and action segmentation tasks, as well as the potential challenges posed by dense prediction tasks.
> > > > > > > > > > >
> > > > > > > > > > > We agree that the primary focus of our method is indeed on point cloud video recognition tasks, as reflected in the title of our paper. Recognition tasks inherently exist information redundancy across point cloud frames, which our one-shot query strategy effectively handles. In contrast, dense prediction tasks, such as action segmentation, require a deeper understanding of motion information for each frame, presenting significantly different challenges.
> > > > > > > > > > >
> > > > > > > > > > > We acknowledge that our method may not bring computational savings in dense prediction tasks and that its performance improvement in such tasks is limited. We believe this distinction between the two task types is an important and valuable topic for the community to explore further. We will incorporate a detailed discussion of these differences and the limitations of our method for dense prediction tasks in the manuscript to provide a balanced and honest assessment,  thereby inspiring researchers to focus more on 4D point cloud videos and the unique challenges posed by dense prediction tasks.
> > > > > > > > > > >
> > > > > > > > > > > Your feedback has inspired us to think more deeply about task definitions in 4D point cloud video research, and we hope this discussion will encourage others to address these challenges and explore innovative solutions. Thank you again for your constructive feedback, and we are grateful for your consideration of our work.

---

> > > > > > > > > > > > ### Comment · Reviewer_Gk9h · 2024-12-03
> > > > > > > > > > > >
> > > > > > > > > > > > I've improved my score to 8, good luck.

---

> > > > > > > > > > > > > ### Author Response · Authors · 2024-12-03
> > > > > > > > > > > > >
> > > > > > > > > > > > > We will incorporate the discussion and improve the paper in the final version. Thank you again for your valuable feedback and best wishes!

---

### Official Review · Reviewer_wjsW · 2024-11-03

**Soundness:** 3
**Presentation:** 3
**Contribution:** 2
**Rating:** 6
**Confidence:** 3

**Summary:**

This paper introduces a novel framework, PvNeXt, designed for efficient point cloud video recognition. PvNeXt leverages a personalized one-shot query mechanism to significantly reduce computational redundancy by avoiding frequent inter-frame queries, commonly seen in traditional methods. The framework incorporates two key modules: the “Motion Imitator” and the “Single-Step Motion Encoder,” which capture temporal dynamics effectively across point cloud sequences. Experimental results show that PvNeXt achieves high accuracy and significantly improved inference speed and memory efficiency on MSR-Action3D and NTU-RGBD datasets.

**Strengths:**

1. Efficiency: PvNeXt uses a one-shot query operation that eliminates complex looping and dense queries, achieving a 23x speedup in inference compared to other methods, which greatly reduces computational costs.

2. Low Resource Usage: The model requires only 0.72M parameters, reducing the parameter count by almost 98% compared to traditional methods, making it suitable for deployment on resource-constrained devices.

3. Innovative Design: The introduction of the “Motion Imitator” and “Single-Step Motion Encoder” modules enables effective temporal and spatial modeling of dynamic information within point cloud sequences.

**Weaknesses:**

1. Lower Accuracy: PvNeXt shows lower recognition accuracy compared to pose [1], RGB [2] and depth-based [2] modalities, highlighting the limitations of using only the point cloud modality for capturing complex dynamic information.

2. Limited Applicability: The model is primarily optimized for action recognition in point cloud videos and lacks validation for other tasks, such as object recognition or scene understanding [3], limiting its generalizability.

3. Potential Difficulty in Capturing Fine-Grained Motion: The one-shot query mechanism, while efficient, might struggle to capture very subtle or intricate motions across frames, which could impact performance in scenarios requiring high precision for small movements.

[1] Chen Y, Zhang Z, Yuan C, et al. Channel-wise topology refinement graph convolution for skeleton-based action recognition[C]//Proceedings of the IEEE/CVF international conference on computer vision. 2021: 13359-13368.

[2] Zhou B, Wang P, Wan J, et al. A unified multimodal de-and re-coupling framework for rgb-d motion recognition[J]. IEEE Transactions on Pattern Analysis and Machine Intelligence, 2023, 45(10): 11428-11442.

[3] Choy C, Gwak J Y, Savarese S. 4d spatio-temporal convnets: Minkowski convolutional neural networks[C]//Proceedings of the IEEE/CVF conference on computer vision and pattern recognition. 2019: 3075-3084.

**Questions:**

1. Could you clarify the practical significance of using point cloud video for action recognition, especially considering that RGB, depth, and pose data have demonstrated better performance?

2. Please update Table 10 to include comparisons with more recent methods, and add efficiency evaluation metrics such as FLOPs and Params.

---

> ### Author Response · Authors · 2024-11-23
> **Response to Reviewer wjsW**
>
> We sincerely appreciate your detailed and insightful reviews. We have provided detailed responses to your comment and updated the relevant content in the revised manuscript, hoping our response can address your concerns.
>
> >**Q1: Clarify the practical significance of using point cloud video for action recognition, especially considering that RGB, depth, and pose data have demonstrated better performance.**
>
> Thanks for your advice! The practical significance of point cloud videos recognition lies in their unique ability to capture the geometric and spatial-temporal structure of 3D environments directly. This capability is particularly critical in applications such as robotics, augmented reality (AR), autonomous navigation, and other tasks requiring an understanding of real-world 3D spatial relationships.
>
> 1) Point cloud videos inherently encode both spatial (3D geometry) and temporal (motion) data, enabling a more accurate depiction of real-world depth, scale, and dynamics. Unlike RGB or depth images, point clouds avoid the distortions or loss of information caused by projections or conversions.
>
> 2) Sensor Compatibility and Real-World Data Trends: With the increasing use of LiDAR and depth sensors, point cloud data is becoming more accessible and practical for real-world applications. Methods leveraging raw point cloud data can directly benefit from this growing trend, bypassing the need for conversion into less efficient formats. While RGB, depth, and pose-based methods currently show superior performance in some contexts, they also have limitations, such as sensitivity to lighting (RGB) or reliance on predefined skeletal models (pose). Point cloud methods, by contrast, offer robustness and adaptability to unstructured environments.
>
> Our goal with this work is not to surpass other modalities in action recognition but to explore and develop methods tailored specifically for point cloud video recognition. By addressing challenges such as computational efficiency and temporal modeling, we aim to contribute to the broader understanding and practical handling of point cloud data, paving the way for its use in future 3D video research and applications.
>
> >**Q2: Limited Applicability and Potential Difficulty in Capturing Fine-Grained Motion.**
>
> Thanks for your advice! It is very necessary to discuss the generalizability of our model.
>
> As you suggested, we extended our evaluation to the HOI4D dataset [1], a large-scale 4D egocentric dataset focusing on category-level human-object interactions. This dataset is particularly challenging due to its diverse action categories, extensive sequences, and detailed human-object interaction dynamics, making it an excellent benchmark for assessing fine-grained motion recognition.
>
> In the action segmentation task of HOI4D, our approach achieved an accuracy of 78.5%, outperforming the advanced method, PPTr, which achieved 77.4%. Notably, our method also demonstrated improvements in additional metrics like Edit Distance and F1 scores (e.g., F1@10 increased from 81.7% to 85.6%, as shown in the table below). These results underline our model's capacity to accurately capture fine-grained motion patterns within egocentric 4D sequences.
>
> We hope this evaluation and explanation address the reviewer’s concerns about the generalizability and fine-grained motion capture capabilities of our model. Thank you for your valuable feedback!
>
> | Method       | Length  | Accuracy | Edit| F1@10 | F1@25 | F1@50 |
> |--------------|---------| ---------| --- |------ |------ |------ |
> |P4Transformer | 150 | 71.2 | 73.1 | 73.8 | 69.2 | 58.2 |
> | PPTr         | 150 | 77.4 | 80.1 | 81.7 | 78.5 | 69.5 |
> | Ours         | 150 | **78.5**	| 84.5 | 85.6 |	82.4 | 73.0 |
>
> Reference:
>
> [1] HOI4D: A 4D Egocentric Dataset for Category-Level Human-Object Interaction CVPR 2022.
>
> >**Q3: Please update Table 10 to include comparisons with more recent methods, and add efficiency evaluation metrics such as FLOPs and Params.**
>
> Thanks for your advice! We have updated Table. 10 as suggested.

---

> > ### Author Response · Authors · 2024-12-04
> >
> > Dear Reviewer wjsW,
> >
> > Thanks again for your great efforts in reviewing this paper!  With the discussion period drawing to a close, we kindly ask if our response has sufficiently addressed your concerns. We put a significant effort into our response, with new experiments and discussions. We sincerely hope you can consider our reply in your assessment.
> >
> >
> > Best regards,
> >
> > Authors

---

### Meta-Review · Area_Chair_AwkE · 2024-12-19

**Metareview:**

This paper received three positive ratings and the reviewers' comments have been adequately addressed. Two reviewers have raised their scores to the positive ones.
In particular, the authors have effectively demonstrated additional experiments to support their claims which have been validated by the reviewers. In addition, the authors also clarified technical details and motivations in the revised version.
Since all the reviewers provide positive ratings, AC recommend to accept this paper.

**Additional Comments On Reviewer Discussion:**

The concerns raised by the reviewers have been well addressed and reviewers all raise their scores.

---

### Decision · Program_Chairs · 2025-01-22

Accept (Poster)